# Performance-Based Human-in-the-Loop Optimal Bipartite Consensus Control for Multi-Agent Systems via Reinforcement Learning

Zongsheng Huang
*School of Automation Engineering*
*University of Electronic Science and Technology of China*
Chengdu 611731, China
zs__Huang@163.com

Tieshan Li
*School of Automation Engineering*
*University of Electronic Science and Technology of China*
Chengdu 611731, China
tieshanli@126.com

Yue Long
*School of Automation Engineering*
*University of Electronic Science and Technology of China*
Chengdu 611731, China
longyue@uestc.edu.cn

Hanqing Yang
*School of Automation Engineering*
*University of Electronic Science and Technology of China*
Chengdu 611731, China
hqyang5517@uestc.edu.cn

*Abstract*—This paper investigates the performance-based human-in-the-loop (HiTL) optimal bipartite consensus control problem for nonlinear multi-agent systems (MASs) under signed topology. First, to respond to any emergencies and guarantee the safety of MASs, the MASs are monitored by human operator sending command signals to the non-autonomous leader. Then, under the joint design architecture of prescribe-time performance function and error transformation, a novel performance index function involving transformed error and control input is developed to achieve optimal bipartite consensus with prescribed-time. Subsequently, the reinforcement learning (RL) method is utilized to learn the solution to Hamilton-Jacobian-Bellman (HJB) equation, in which the fuzzy logic systems (FLSs) are employed to implement the method. Finally, the simulation results depict the effectiveness of the constructed control scheme.

*Index Terms*—Human-in-the-loop control, prescribed-time control, reinforcement learning, nonlinear multi-agent systems.

## I. Introduction

In recent years, with the rapid development of multiple unmanned aerial vehicles (UAVs) [1], multiple unmanned ground vehicles (UGVs) [2] and other fields, multi-agent systems (MASs) have been paid more and more attention by scholars. As one of the hot issues in control problems of MASs, consensus control problems have been widely studied. As a branch of consensus control, bipartite consensus was first introduced in [3] taking both competition and cooperation relationships between agents into consideration. For bipartite consensus, the agents eventually converge to two states of opposite sign but equal size. In [4]-[6], the various control strategies of bipartite consensus have been designed broadly.

This work was supported in part by the National Natural Science Foundation of China under Grant 51939001, Grant 62273072, and Grant 62203088, in part by the Natural Science Foundation of Sichuan Province under Grant 2022NSFSC0903.(*Corresponding author: Tieshan Li*)

Notably, the MASs mentioned above are fully autonomous. However, incidents with Boeing 737 jetliners and Tesla's autonomous driving systems have raised serious concerns and highlighted the challenges that fully autonomous MASs face in making judgments during in uncertain and complex environments. Therefore, it is urgent to develop monitoring schemes to complete tasks when MASs encounter unexpected situations [7]. Fortunately, the human-in-the-loop (HiTL) control approach was introduced in MASs to supervise the entire system to respond to sudden changes by sending commands to the leader agent [8]. Later, many studies on HiTL control for MASs have emerged in [9]-[15]. In [9], the HiTL formation tracking control scheme together with edge-based event-driven mechanism was constructed for MASs. Considering stochastic actuation attacks, in [13], the prescribed-time and prescribed-accuracy HiTL cluster consensus control problem has been solved. In view of the ability to deal with emergencies, the HiTL control approach has also been favored by multi-UAV systems in [14], [15].

Optimal control, a widely used control method, has garnered significant attention. For nonlinear systems, the optimal solution is derived from the Hamilton-Jacobian-Bellman (HJB) equation. However, obtaining the solution of HJB equation through numerical methods is infeasible. To overcome this challenge, reinforcement learning (RL) that motivated by animal behaviors was proposed as a powerful tool [16]. The core idea of RL is to approximate the solution of the HJB equation using a function approximation structure. The value iteration algorithm, one of the valuable algorithms in RL, was developed by Murray *et al.* in [17], in which the convergence analysis was also detailed. In [18], the policy iteration algorithm, as another equally important algorithm, was designed to obtain the optimal saturation controller for

nonlinear systems. Based on the previous work, RL method has been used to solve the optimal problem for MASs. In [19], an optimal control protocol based on RL was designed to achieve containment control without prior knowledge of the system dynamics. For unknown discrete-time MASs, in [20], the optimal bipartite consensus control problem was solved. Nevertheless, the above results only conclude that the optimal controller is globally asymptotically stable. It is important to note that achieving specified accuracy within a given time is crucial in many fields.

Fortunately, the prescribed-time control (PTC) was firstly proposed by Song *et al.* [21]. The PTC distinguishes from finite-time control and fixed-time control, in which the preset settling time is not related to the initial values of the system. Depending on [21], in [22], the convergence rate can be pre-determined as needed, and a general method for constructing the time-varying rate function was provided. In [23], a novel time-varying constraint function was devised to guarantee that the system remains operational beyond the prescribed time, leading to a global result. In particular, the PTC-based HiTL control scheme was developed to realize the cluster consensus within given time in [13]. However, to the best of the authors' knowledge, the bipartite consensus control scheme considering both optimal performance and prescribed-time performance under the framework of HiTL control has not been fully explored, which promotes our research.

Driven by these observations, this paper focuses on investigating the performance-based HiTL optimal bipartite consensus control problem. The main contributions are summarized below.

(1) Unlike the autonomous leader described in [4]-[6] which lacked intelligent decision-making, this paper aims to improve the security, stability, and emergency response capabilities of the system by designing the leader of the MASs to be non-autonomous, where the time-varying control input is governed by a human operator.

(2) Compared with the existing optimal results for MASs in [19], [20], to realize both optimal performance and prescribed-time performance, a unified design framework of PTC and RL method is proposed, where the settling time and accuracy can be preset without initial values.

The structure is given below. In Section II, the considered system and some assumptions are given. In Section III, the main results including the PTC performance function and optimal controller are designed. In Section IV, the convergence analysis is provided. The simulation results is given in Section V. Finally, the conclusion is presented in Section VI.

## II. PROBLEM FORMULATION AND PRELIMINARIES

### A. Signed Communication Topologies

The structurally balanced bipartition communication topology containing $N$ followers is represented by a directed graph $\mathcal{G} = \{\mathcal{V}, \varepsilon, \mathcal{A}\}$, where $\mathcal{V} = \{\mathcal{V}_1, \mathcal{V}_2, \cdots, \mathcal{V}_N\}$ represents the vertex set, which is divided into the cooperative set $\mathcal{V}_\alpha$ and competitive set $\mathcal{V}_\beta$ such that $\mathcal{V}_\alpha \cap \mathcal{V}_\beta = 0$ and $\mathcal{V}_\alpha \cup \mathcal{V}_\beta = \mathcal{V}$.

$\varepsilon \subseteq \mathcal{V} \times \mathcal{V}$ represents the edge set of $N$ followers. Let $\mathcal{A} = [a_{ij}] \in \mathbb{R}^{N \times N}$ be the signed weight matrix, where $a_{ij} > 0$ if $(\mathcal{V}_i, \mathcal{V}_j) \in \mathcal{V}_m$, $m \in \{\alpha, \beta\}$ and $a_{ij} < 0$ if $\mathcal{V}_i \in \mathcal{V}_m, \mathcal{V}_j \in \mathcal{V}_n, m \neq n, m, n \in \{\alpha, \beta\}$. The neighbor set of $i$th follower is defined as $\mathcal{N}_i = \{j \in \mathcal{V} : a_{ij} \neq 0\}$. Define $\mathcal{L} = \mathcal{D} - \mathcal{A} \in \mathbb{R}^{N \times N}$ as the Laplacian matrix of $\mathcal{G}$, where $\mathcal{D} = \text{diag}(d_1, d_2, \cdots, d_N) \in \mathbb{R}^{N \times N}$ denotes the degree matrix with $d_i = \sum_{j=1}^{N} |a_{ij}|$.

The argument graph consisting of one leader and $N$ followers is denoted as $\tilde{\mathcal{G}} = \{\tilde{\mathcal{V}}, \tilde{\varepsilon}\}$, in which $\tilde{\mathcal{V}} = \{\mathcal{V}_0, \mathcal{V}_1, \mathcal{V}_2, \cdots, \mathcal{V}_N\}$ and $\tilde{\varepsilon} \subseteq \tilde{\mathcal{V}} \times \tilde{\mathcal{V}}$. Let $\mathcal{B} = \text{diag}\{|b_1|, |b_2|, \cdots, |b_N|\} \in \mathbb{R}^{N \times N}$, where $b_i = 1$ indicates that the information of the leader is available for the $i$th node and $b_i > 0$ represents cooperative relation, $b_i < 0$ represents competitive relation.

### B. Problem Formulation

Assume that the nonlinear MAS is composed of $N(\geq 2)$ followers and one leader. The dynamics model of $i$th follower is provided as

$$\dot{x}_i = f_i(x_i) + g_i(x_i)u_i, i = 1, 2, \cdots, N \qquad (1)$$

where $x_i(t) \in \mathbb{R}^n$ denotes state, $u_i(t) \in \mathbb{R}^m$ is control input, $f_i(x_i) \in \mathbb{R}^n$ is internal dynamics and $g_i(x_i) \in \mathbb{R}^{n \times m}$ is input dynamics.

Next, the dynamics of the human-manipulated leader is given as

$$\dot{x}_0^h = f_0^h(x_0^h) + u_0^h, \qquad (2)$$

where $x_0^h(t) \in \mathbb{R}^n$ denotes state and $u_0^h(t) \in \mathbb{R}^m$ is nonzero control input of human operator sending to leader, $f_0^h(x_0^h) \in \mathbb{R}^n$ represents internal dynamics.

The following assumptions and lemma are imposed.

**Assumption 1.** *[19] The signed graph $\mathcal{G}$ has a directed spanning tree.*

**Assumption 2.** *[24] The input of human operator always makes the leader (2) stable.*

**Lemma 1.** *[25]: The FLS can estimate a nonlinear continuous function $f(\mathfrak{r}) \in \mathbb{R}$ on a compact set $\Omega_f \in \mathbb{R}^n$ as*

$$\sup_{\mathfrak{r} \in \Omega_f} |f(\mathfrak{r}) - \Theta^T \phi(\mathfrak{r})| \leq b \qquad (3)$$

*with $b > 0$.*

## III. MAIN RESULTS

### A. Prescribed-Time Function and Error Transformation

To achieve prescribed-time (PT) performance for MASs, the PT performance function $\vartheta(t)$ is given as

$$\vartheta(t) = \begin{cases} \iota e^{-\beta(\frac{T}{T-t})^h} + \vartheta_{T_r}, & 0 < t < T_r \\ \vartheta_{T_r}, & t \geq T_r \end{cases} \qquad (4)$$

where $h > 0, \iota > 0, \beta > 0, \vartheta_{T_r} > 0, 0 < T_r < \infty$ and $0 < \vartheta_{T_r} < \infty$ represent the user-defined settling time and steady-state tracking accuracy, respectively.

Construct the bipartite consensus error as $e_i = \sum_{j=1}^{N} |a_{ij}|(x_i - \text{sign}(a_{ij})x_j) + |b_i|(x_i - \text{sign}(b_i)x_0^h)$, $e_i = [e_{i,1}, \cdots, e_{i,n}]^T \in \mathbb{R}^n$ and adopt the error transformation function as

$$\varrho_{i,\imath} = \tan(\frac{\pi}{2}\frac{e_{i,\imath}}{\vartheta}), \imath = 1, \cdots, n, \quad (5)$$

where $|e_{i,\imath}(0)| < \vartheta(0)$.

Based on (5), it yields

$$e_{i,\imath} = \frac{2\vartheta}{\pi}\arctan(\varrho_{i,\imath}), \imath = 1, \cdots, n, i = 1, \cdots, N. \quad (6)$$

**Remark 1.** *From (5), the inequality $-\vartheta \le e_{i,\imath} \le \vartheta, \forall t \ge 0$ holds. Combined the definition in (4), it further observes that $-\vartheta_{T_r} \le e_{i,\imath} \le \vartheta_{T_r}, \forall t \ge T_r$ if $\varrho_{i,\imath}$ is bounded, which means the PT performance of $e_i$ can be ensured.*

### B. Optimal control Scheme Design

Define the performance index function as

$$
\begin{aligned}
J_i &= \int_t^\infty (e_i^T \mathcal{Q}_i e_i + u_i^T \mathcal{R}_i u_i) d\tau \\
&= \int_t^\infty ((\frac{2\vartheta}{\pi}\mathscr{A}_i)^T \mathcal{Q}_i (\frac{2\vartheta}{\pi}\mathscr{A}_i) + u_i^T \mathcal{R}_i u_i) d\tau,
\end{aligned} \quad (7)
$$

where $\mathcal{Q}_i$ and $\mathcal{R}_i$ are symmetric positive definite matrices with suitable dimensions, $\mathscr{A}_i = [\mathscr{A}_{i,1}, \cdots, \mathscr{A}_{i,n}]^T = [\arctan(\varrho_{i,1}), \cdots, \arctan(\varrho_{i,n})]^T$.

Taking the time derivative of $\mathscr{A}_{i,\imath}$, one has

$$\dot{\mathscr{A}}_{i,\imath} = \frac{1}{1 + \varrho_{i,\imath}^2}\chi_{i,\imath}(\dot{e}_{i,\imath} - \nu_{i,\imath}), \quad (8)$$

where $\chi_{i,\imath} = \frac{\pi}{2\vartheta\cos^2(\frac{\pi}{2}\frac{e_{i,\imath}}{\vartheta})}$, $\nu_{i,\imath} = \frac{e_{i,\imath}\dot{\vartheta}}{\vartheta}$, $\dot{e}_i = \Gamma_i(f_i + g_i u_i) - \sum_{j=1}^N a_{ij}\dot{x}_j - b_i \dot{x}_0^h$ and $\Gamma_i = d_i + |b_i|$.

Then, define the Hamiltonian function as

$$
\begin{aligned}
H_i(\mathscr{A}_i, \vartheta, u_i, \frac{\partial J_i}{\partial \mathscr{A}_i}, \frac{\partial J_i}{\partial \vartheta}) &= (\frac{2\vartheta}{\pi}\mathscr{A}_i)^T \mathcal{Q}_i (\frac{2\vartheta}{\pi}\mathscr{A}_i) \\
&+ u_i^T \mathcal{R}_i u_i + \frac{\partial J_i}{\partial \mathscr{A}_i}[\bar{\chi}_i(\dot{e}_i - \nu_i)] + \frac{\partial J_i}{\partial \vartheta}\frac{\partial \vartheta}{\partial t} \\
&= (\frac{2\vartheta}{\pi}\mathscr{A}_i)^T \mathcal{Q}_i (\frac{2\vartheta}{\pi}\mathscr{A}_i) + u_i^T \mathcal{R}_i u_i + \frac{\partial J_i}{\partial \varrho_i}[\chi_i(\dot{e}_i - \nu_i)] \\
&+ \frac{\partial J_i}{\partial \vartheta}\frac{\partial \vartheta}{\partial t},
\end{aligned} \quad (9)
$$

where $\bar{\chi}_i = \text{diag}\{\frac{\chi_{i,1}}{1+\varrho_{i,1}^2}, \cdots, \frac{\chi_{i,n}}{1+\varrho_{i,n}^2}\}$, $\nu_i = [\nu_{i,1}, \cdots, \nu_{i,n}]$ and $\chi_i = \text{diag}\{\chi_{i,1}, \cdots, \chi_{i,n}\}$.

The corresponding HJB equation is given as

$$\min_{u_i} H_i(\mathscr{A}_i, \vartheta, u_i^*, \frac{\partial J_i^*}{\partial \mathscr{A}_i}, \frac{\partial J_i^*}{\partial \vartheta}) = 0. \quad (10)$$

Differentiating the (10) with respect to $u_i$, one has

$$u_i^* = -\frac{\Gamma_i}{2}\mathcal{R}_i^{-1}g_i^T \chi_i^T \frac{\partial J_i^*}{\partial \varrho_i}. \quad (11)$$

Substituting (11) into (10), (10) becomes

$$
\begin{aligned}
&(\frac{2\vartheta}{\pi}\mathscr{A}_i)^T \mathcal{Q}_i (\frac{2\vartheta}{\pi}\mathscr{A}_i) + \frac{\partial J_i^*}{\partial \varrho_i}[\chi_i(\Gamma_i f_i - \sum_{j=1}^N a_{ij}\dot{x}_i - b_i \dot{x}_0^h \\
&- \nu_i)] + \frac{\partial J_i^*}{\partial \vartheta}\frac{\partial \vartheta}{\partial t} - \frac{\Gamma_i^2}{4}\frac{\partial J_i^*}{\partial \varrho_i^T}g_i \chi_i \mathcal{R}_i^{-1}\chi_i^T g_i^T \frac{\partial J_i^*}{\partial \varrho_i} = 0.
\end{aligned}
$$

Inspired by [26], $\frac{\partial J_i^*}{\partial \varrho_i}$ can be segmented as

$$\frac{\partial J_i^*}{\partial \varrho_i} = \frac{2k_i}{\Gamma_i}\chi_i^{-2}\varrho_i + \frac{2}{\Gamma_i}\chi_i^{-2}\mathcal{F}_i(\mathscr{X}_i) + \frac{1}{\Gamma_i}\chi_i^{-2}\mathcal{J}_i(\mathcal{X}_i), \quad (12)$$

where $k_i > 0$, $\mathcal{F}_i(\mathscr{X}_i) = \mathcal{R}_i \chi_i(f_i(x_i) - \dot{x}_0^h - o^{-1}\nu_i)$ with $o = \lambda_{\max}(\mathcal{L} + \mathcal{B})$, $\mathcal{J}_i(\mathcal{X}_i) = -2k_i\varrho_i^2 - 2\mathcal{F}_i(\mathscr{X}_i) + k_i\chi_i^2 \frac{\partial J_i^*}{\partial \varrho_i}$.

Substituting (12) into (11), one has

$$
\begin{aligned}
u_i^* = &- k_i \mathcal{R}_i^{-1}\chi_i^{-1}\varrho_i - \mathcal{R}_i^{-1}\chi_i^{-1}\mathcal{F}_i(\mathscr{X}_i) \\
&- \frac{1}{2}\mathcal{R}_i^{-1}\chi_i^{-1}\mathcal{J}_i(\mathcal{X}_i).
\end{aligned} \quad (13)
$$

### C. PI Algorithm and FLSs-Based Implementation

Obviously, the HJB equation can not be acquired by numerical methods. Therefore, the PI approach is given in Algorithm 1 to find the optimal result.

---

**Algorithm 1:** PI Algorithm for Solving PT Optimal Consensus Control Policy

---

**1** *Step 1:* Initialization. Give an initial control protocols $u_i^{(0)}, \forall i$.

**2** *Step 2:* Policy evaluation. Solve the cost function $J_i^l$ as: $H_i(\mathscr{A}_i, \vartheta, u_i^*, \frac{\partial J_i^l}{\partial \mathscr{A}_i}, \frac{\partial J_i^l}{\partial \vartheta}) = 0$.

**3** *Step 3:* Policy improvement. Update optimal control input $u_i^{(l+1)}$ as Eq. (13).

**4** *Step 4:* If $\|J_i^{(l+1)} - J_i^{(l)}\| \le \aleph$ with the predefined parameter $\aleph > 0$, stop; otherwise, set $l = l + 1$ and return to Step 2.

---

The convergence and optimality of Algorithm 1 have been proved in [27] and are omitted here.

In view of the unknown term $\mathcal{F}_i(\mathscr{X}_i)$ and $\mathcal{J}_i(\mathcal{X}_i)$ in (13), the FLSs is used to approximate these terms as.

$$\mathcal{F}_i(\mathscr{X}_i) = \omega_{\mathcal{F}_i}^T \phi_{\mathcal{F}_i}(\mathscr{X}_i) + \epsilon_{\mathcal{F}_i}(\mathscr{X}_i), \quad (14)$$

$$\mathcal{J}_i(\mathcal{X}_i) = \omega_{\mathcal{J}_i}^T \phi_{\mathcal{J}_i}(\mathscr{X}_i) + \epsilon_{\mathcal{J}_i}(\mathcal{X}_i), \quad (15)$$

where $\omega_{\mathcal{F}_i} \in \mathbb{R}^{h_{c1} \times n}$ and $\omega_{\mathcal{J}_i} \in \mathbb{R}^{h_{c2} \times n}$ represent ideal weight matrices with $h_{c1}$ and $h_{c2}$ are the number of fuzzy rules; $\phi_{\mathcal{F}_i} \in \mathbb{R}^{h_{c1}}$ and $\phi_{\mathcal{J}_i} \in \mathbb{R}^{h_{c2}}$ are fuzzy basis functions; $\epsilon_{\mathcal{F}_i}(\mathscr{X}_i)$ and $\epsilon_{\mathcal{J}_i}(\mathcal{X}_i)$ denote bounded approximation errors.

Thus, (13) becomes

$$
\begin{aligned}
u_i^* = &- k_i \mathcal{R}_i^{-1}\chi_i^{-1}\varrho_i - \mathcal{R}_i^{-1}\chi_i^{-1}(\omega_{\mathcal{F}_i}^T \phi_{\mathcal{F}_i}(\mathscr{X}_i) + \epsilon_{\mathcal{F}_i}(\mathscr{X}_i)) \\
&- \frac{1}{2}\mathcal{R}_i^{-1}\chi_i^{-1}(\omega_{\mathcal{J}_i}^T \phi_{\mathcal{J}_i}(\mathcal{X}_i) + \epsilon_{\mathcal{J}_i}(\mathcal{X}_i)).
\end{aligned}
$$

However, the $\omega_{\mathcal{F}_i}$ and $\omega_{\mathcal{J}_i}$ are unknown, the estimation forms of (14) and (15) are

$$\hat{\mathcal{F}}_i(\mathscr{X}_i) = \hat{\omega}_{\mathcal{F}_i}^T \phi_{\mathcal{F}_i}(\mathscr{X}_i), \quad (16)$$

$$\hat{\mathcal{J}}_i(\mathcal{X}_i) = \hat{\omega}_{\mathcal{J}_i}^T \phi_{\mathcal{J}_i}(\mathcal{X}_i), \tag{17}$$

where $\hat{\omega}_{\mathcal{F}_i} \in \mathbb{R}^{h_{c1} \times n}$ and $\hat{\omega}_{\mathcal{J}_i} \in \mathbb{R}^{h_{c2} \times n}$ represent estimated weight matrices.

According to (16) and (17), one has

$$\hat{u}_i^* = -k_i \mathcal{R}_i^{-1} \chi_i^{-1} \varrho_i - \mathcal{R}_i^{-1} \chi_i^{-1}(\hat{\omega}_{\mathcal{F}_i}^T \phi_{\mathcal{F}_i}(\mathcal{X}_i)) \\ - \frac{1}{2}\mathcal{R}_i^{-1}\chi_i^{-1}(\hat{\omega}_{\mathcal{J}_i}^T \phi_{\mathcal{J}_i}(\mathcal{X}_i)). \tag{18}$$

The updating laws are constructed as

$$\dot{\hat{\omega}}_{\mathcal{F}_i} = \mathscr{C}_i(o\phi_{\mathcal{F}_i}(\mathcal{X}_i)\varrho_i^T \mathcal{R}_i^{-1} - r_{\mathcal{F}_i}\hat{\omega}_{\mathcal{F}_i}), \tag{19}$$

$$\dot{\hat{\omega}}_{\mathcal{J}_i} = -r_{\mathcal{J}_i}(\phi_{\mathcal{J}_i}^T(\mathcal{X}_i)\phi_{\mathcal{J}_i}(\mathcal{X}_i) + r\mathcal{I}_{h_{c2}})\hat{\omega}_{\mathcal{J}_i}, \tag{20}$$

where $\mathscr{C}_i \in \mathbb{R}^{h_{c1} \times h_{c1}}$ is a positive-definite matrix, $r_{\mathcal{F}_i} > 0, r_{\mathcal{J}_i} > 0, r > 0$ are design parameters.

## IV. STABILITY ANALYSIS

**Theorem 1.** *Consider the MAS consisting of followers (1) and leader (1) under Assumption 1-3, by choosing $k_i > \frac{3}{4}$ and adopting optimal control input (18) and adaptive law (19) and (20), then the consensus error can converge to the prescribed accuracy within prescribed time.*

*Proof.* Develop the Lyapunov function as

$$V = \frac{1}{2}\varrho^T \varrho + \frac{1}{2}\sum_{j=1}^N (\tilde{\omega}_{\mathcal{F}_i}^T \mathscr{C}_i^{-1}\tilde{\omega}_{\mathcal{F}_i} + \tilde{\omega}_{\mathcal{J}_i}^T \tilde{\omega}_{\mathcal{J}_i}) \tag{21}$$

where $\varrho = [\varrho_1^T, \cdots, \varrho_n^T]^T \in \mathbb{R}^{N \times n}$, estimation error $\tilde{\omega}_{\mathcal{F}_i} = \omega_{\mathcal{F}_i} - \hat{\omega}_{\mathcal{F}_i}$ and $\tilde{\omega}_{\mathcal{J}_i} = \omega_{\mathcal{J}_i} - \hat{\omega}_{\mathcal{J}_i}$.

Invoking (5), (19) and (20), it yields

$$\dot{V} = \varrho^T[\chi(\mathcal{L} + \mathcal{B})\dot{e} - \chi\nu] - \sum_{j=1}^N(\tilde{\omega}_{\mathcal{F}_i}^T(o\phi_{\mathcal{F}_i}(\mathcal{X}_i)\varrho_i^T \mathcal{R}_i^{-1}$$

$$- r_{\mathcal{F}_i}\hat{\omega}_{\mathcal{F}_i}) + \sum_{j=1}^N(\tilde{\omega}_{\mathcal{J}_i}^T(r_{\mathcal{J}_i}(\phi_{\mathcal{J}_i}^T(\mathcal{X}_i)\phi_{\mathcal{J}_i}(\mathcal{X}_i) + r)\mathcal{I}_{h_{c2}})\hat{\omega}_{\mathcal{J}_i})$$

$$\leq \sum_{j=1}^N \varrho_i^T o(-k_i \mathcal{R}_i^{-1}\varrho_i - \mathcal{R}_i^{-1}\tilde{\omega}_{\mathcal{F}_i}^T \phi_{\mathcal{F}_i}(\mathcal{X}_i) + \mathcal{R}_i^{-1}\epsilon_{\mathcal{F}_i}(\mathcal{X}_i)$$

$$- \frac{1}{2}\mathcal{R}_i^{-1}\hat{\omega}_{\mathcal{J}_i}^T \phi_{\mathcal{J}_i}(\mathcal{X}_i)) - \sum_{j=1}^N(\tilde{\omega}_{\mathcal{F}_i}^T(o\phi_{\mathcal{F}_i}(\mathcal{X}_i)\varrho_i^T \mathcal{R}_i^{-1}$$

$$- r_{\mathcal{F}_i}\hat{\omega}_{\mathcal{F}_i}) + \sum_{j=1}^N(\tilde{\omega}_{\mathcal{J}_i}^T(r_{\mathcal{J}_i}(\phi_{\mathcal{J}_i}^T(\mathcal{X}_i)\phi_{\mathcal{J}_i}(\mathcal{X}_i) + r)\mathcal{I}_{h_{c2}})\hat{\omega}_{\mathcal{J}_i})$$

$$\leq \sum_{j=1}^N \varrho_i^T o(-k_i \mathcal{R}_i^{-1}\varrho_i + \mathcal{R}_i^{-1}\epsilon_{\mathcal{F}_i}(\mathcal{X}_i) - \frac{\mathcal{R}_i^{-1}}{2}\hat{\omega}_{\mathcal{J}_i}^T \phi_{\mathcal{J}_i}(\mathcal{X}_i))$$

$$+ \sum_{j=1}^N(r_{\mathcal{F}_i}\tilde{\omega}_{\mathcal{F}_i}^T \hat{\omega}_{\mathcal{F}_i}) + \sum_{j=1}^N(\tilde{\omega}_{\mathcal{J}_i}^T(r_{\mathcal{J}_i}(\phi_{\mathcal{J}_i}^T(\mathcal{X}_i)\phi_{\mathcal{J}_i}(\mathcal{X}_i)$$

$$+ r)\mathcal{I}_{h_{c2}})\hat{\omega}_{\mathcal{J}_i}). \tag{22}$$

Using Young's inequality, we have

$$o\varrho_i^T \mathcal{R}_i^{-1}\epsilon_{\mathcal{F}_i} \leq \frac{o}{2}\mathcal{R}_i^{-1}||\varrho_i||^2 + \frac{o}{2}\mathcal{R}_i^{-1}||\epsilon_{\mathcal{F}_i}||^2, \tag{23}$$

$$-\frac{o\mathcal{R}_i^{-1}}{2}\hat{\omega}_{\mathcal{J}_i}^T \phi_{\mathcal{J}_i}(\mathcal{X}_i)) \leq \frac{o\mathcal{R}_i^{-1}}{4}\hat{\omega}_{\mathcal{J}_i}^T \phi_{\mathcal{J}_i}(\mathcal{X}_i))\phi_{\mathcal{J}_i}^T(\mathcal{X}_i))\hat{\omega}_{\mathcal{J}_i}$$

$$+ \frac{o\mathcal{R}_i^{-1}}{4}||\varrho_i||^2, \tag{24}$$

$$\tilde{\omega}_{\mathcal{F}_i}^T \hat{\omega}_{\mathcal{F}_i} \leq -\frac{1}{2}\tilde{\omega}_{\mathcal{F}_i}^T \tilde{\omega}_{\mathcal{F}_i} + \frac{1}{2}\omega_{\mathcal{F}_i}^T \omega_{\mathcal{F}_i}, \tag{25}$$

$$\tilde{\omega}_{\mathcal{J}_i}^T(\phi_{\mathcal{J}_i}^T(\mathcal{X}_i)\phi_{\mathcal{J}_i}(\mathcal{X}_i) + r\mathcal{I}_{h_{c2}})\hat{\omega}_{\mathcal{J}_i} \leq \frac{-\tilde{\omega}_{\mathcal{J}_i}^T}{2}(\phi_{\mathcal{J}_i}^T(\mathcal{X}_i)\phi_{\mathcal{J}_i}(\mathcal{X}_i)$$

$$+ r\mathcal{I}_{h_{c2}})\tilde{\omega}_{\mathcal{J}_i} + \frac{\hat{\omega}_{\mathcal{J}_i}^T}{2}(\phi_{\mathcal{J}_i}^T(\mathcal{X}_i)\phi_{\mathcal{J}_i}(\mathcal{X}_i) + r\mathcal{I}_{h_{c2}})\hat{\omega}_{\mathcal{J}_i}. \tag{26}$$

Calculating (22) by bringing (23)-(26), one has

$$\dot{V} \leq -\sum_{j=1}^N o\mathcal{R}_i^{-1}(k_i - \frac{3}{4})||\varrho_i||^2 - \sum_{j=1}^N \frac{r_{\mathcal{F}_i}}{2}\tilde{\omega}_{\mathcal{F}_i}^T \tilde{\omega}_{\mathcal{F}_i}$$

$$- \sum_{j=1}^N(\frac{\tilde{\omega}_{\mathcal{J}_i}^T}{2}(\phi_{\mathcal{J}_i}^T(\mathcal{X}_i)\phi_{\mathcal{J}_i}(\mathcal{X}_i) + r\mathcal{I}_{h_{c2}})\tilde{\omega}_{\mathcal{J}_i}) + \Lambda$$

$$\leq -\frac{\kappa_1}{2}\sum_{j=1}^N ||\varrho_i||^2 - \frac{\kappa_2}{2}\sum_{j=1}^N \tilde{\omega}_{\mathcal{F}_i}^T \mathscr{C}_i^{-1}\tilde{\omega}_{\mathcal{F}_i} - \frac{\kappa_3}{2}\tilde{\omega}_{\mathcal{J}_i}^T \tilde{\omega}_{\mathcal{J}_i}$$

$$+ \Lambda$$

$$\leq -\kappa V + \Lambda, \tag{27}$$

where $\Lambda = \sum_{j=1}^N \frac{o}{2}\mathcal{R}_i^{-1}||\epsilon_{\mathcal{F}_i}||^2 + \sum_{j=1}^N \frac{o\mathcal{R}_i^{-1}}{4}\hat{\omega}_{\mathcal{J}_i}^T \phi_{\mathcal{J}_i}(\mathcal{X}_i))\phi_{\mathcal{J}_i}^T(\mathcal{X}_i))\hat{\omega}_{\mathcal{J}_i} + \sum_{j=1}^N \frac{o\mathcal{R}_i^{-1}}{4}||\varrho_i||^2 + \sum_{j=1}^N \frac{r_{\mathcal{F}_i}}{2}\omega_{\mathcal{F}_i}^T \omega_{\mathcal{F}_i} + \sum_{j=1}^N \frac{\hat{\omega}_{\mathcal{J}_i}^T}{2}(\phi_{\mathcal{J}_i}^T(\mathcal{X}_i)\phi_{\mathcal{J}_i}(\mathcal{X}_i) + r\mathcal{I}_{h_{c2}})\hat{\omega}_{\mathcal{J}_i}$, $\kappa_1 = \min_{i=1,\cdots,N}\{2o\mathcal{R}_i^{-1}(k_i - \frac{3}{4})\}$, $\kappa_2 = \min_{i=1,\cdots,N}\{\frac{r_{\mathcal{F}_i}}{\lambda_{\max}(\mathscr{C}_i^{-1})}\}$, $\kappa_3 = \min_{i=1,\cdots,N}\{r_{\mathcal{J}_i}\lambda_{\min}(\phi_i)\}$, $\kappa = \min\{\kappa_1, \kappa_2, \kappa_3\}$, $\lambda_{\min}(\phi_i)$ is the minimal eigenvalue of $\phi_{\mathcal{J}_i}^T(\mathcal{X}_i)\phi_{\mathcal{J}_i}(\mathcal{X}_i)$. $\qquad\square$

## V. SIMULATION

A nonlinear MAS composed by four single-link robot arms (three followers and one human-controlled leader) is given to verify the effectiveness of the proposed control scheme. The model of agent is given as [12]

$$J_i\ddot{q}_i + D_i\dot{q}_i + M_i gd_i \sin(q_i) = u_i, i = 1, \cdots, 3,$$

the physical parameters of $g, M_i, D_i, J_i$ and $d_i$ can be found in [12] for details. $u_0^h$ is set as

$$u_0^h = \begin{cases} 0.3 * \sin(t) * \sin(t), 0 \leq t < 15 \\ 0, 15 \leq t < 30 \\ \sin(t) * \cos(t), 30 \leq t \leq 50. \end{cases}$$

The communication graph is shown below

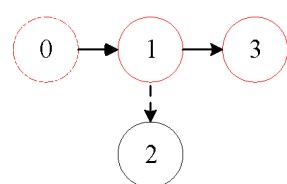

Fig. 1: Communication graph.

As shown in Fig. 1, it can be obtained that

$$\mathcal{A} = \begin{bmatrix} 0 & -1 & 1 \\ 0 & 0 & 0 \\ 0 & 0 & 0 \end{bmatrix}, \mathcal{L} = \begin{bmatrix} 2 & 1 & -1 \\ 0 & 0 & 0 \\ 0 & 0 & 0 \end{bmatrix},$$

$\mathcal{B} = \operatorname{diag}\{1,0,0,0,0\}$.

For PT performance function, select $\vartheta_{T_r} = 0.06, T_r = 3s$. The initial state values of followers and leader are presented in Table 1.

TABLE I: Initial state values of followers and leader.

| State | $i=0$ | $i=1$ | $i=2$ | $i=3$ |
|-------|-------|-------|-------|-------|
| $x_{i,1}(0)$ | 1 | 0.8 | 0.5 | 0.8 |
| $x_{i,2}(0)$ | -1 | 0.8 | -0.5 | -0.8 |

For the unknown term $\mathcal{F}_i(\mathscr{X}_i)$, $\mathscr{X}_i = [x_i, x_0^h, \dot{x}_0^h, \vartheta, \dot{\vartheta}]^T$ and defined over $[-6,6]$. Choose $\mathscr{X}_i^0 = [[-6-\mathscr{L}, -6+\mathscr{L}]^T, \cdots, [-6-\mathscr{L}, -6+\mathscr{L}]^T]^T$ and
$$\underbrace{}_{5}$$
$\phi_{\mathcal{F}_i^{\mathscr{L}}}(\mathscr{X}_i) = \exp(-\frac{(\mathscr{X}_i - \mathscr{X}_i^0)^T(\mathscr{X}_i - \mathscr{X}_i^0)}{2})$.

For the unknown term $\mathcal{J}_i(\mathscr{X}_i)$, $\mathscr{X}_i = [x_i, \varrho_i, x_0^h, \dot{x}_0^h, \vartheta, \dot{\vartheta}]^T$ and defined over $[-6,6]$. Choose $\mathscr{X}_i^0 = [[-6-\mathscr{L}, -6+\mathscr{L}]^T, \cdots, [-6-\mathscr{L}, -6+\mathscr{L}]^T]^T$ and
$$\underbrace{}_{6}$$
$\phi_{\mathcal{J}_i}(\mathscr{X}_i)^{\mathscr{L}}(\mathscr{X}_i) = \exp(-\frac{(\mathscr{X}_i - \mathscr{X}_i^0)^T(\mathscr{X}_i - \mathscr{X}_i^0)}{2})$.

For updating law (19) and (20), $\hat{\omega}_{\mathcal{F}_1}(0) = \hat{\omega}_{\mathcal{F}_2}(0) = \hat{\omega}_{\mathcal{F}_3}(0) = [0.1]_{12\times2}$, $\hat{\omega}_{\mathcal{J}_1}(0) = \hat{\omega}_{\mathcal{J}_2}(0) = \hat{\omega}_{\mathcal{J}_3}(0) = [0.92]_{12\times2}$, $\mathscr{C}_1 = \operatorname{diag}\underbrace{\{0.5, \cdots, 0.5\}}_{12}, \mathscr{C}_2 = \operatorname{diag}\underbrace{\{0.7, \cdots, 0.7\}}_{12}, \mathscr{C}_3 = \operatorname{diag}\underbrace{\{0.3, \cdots, 0.3\}}_{12}$, $\mathscr{R}_i = \operatorname{diag}\{0.8, 0.8\}, r_{\mathcal{F}_i} = 2, k_i = 45, r_{\mathcal{J}_i} = 1$.

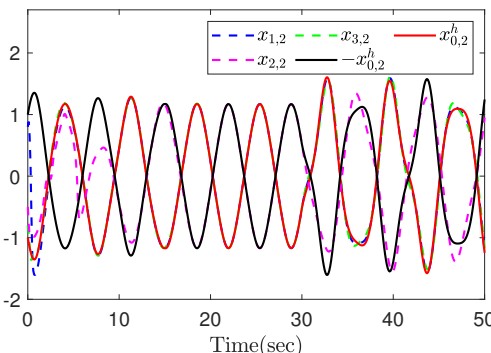

Fig. 3: Curves of $\tilde{x}_{i,2}$, $x_{0,2}^h$ and $-x_{0,2}^h$.

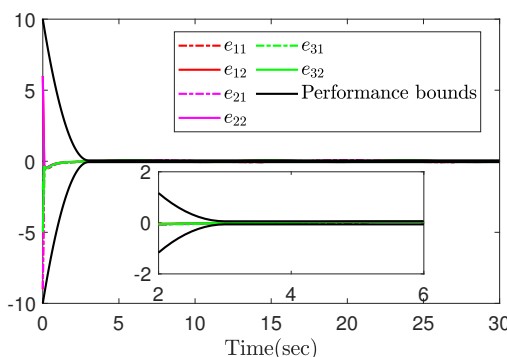

Fig. 4: Curves of errors and performance bounds.

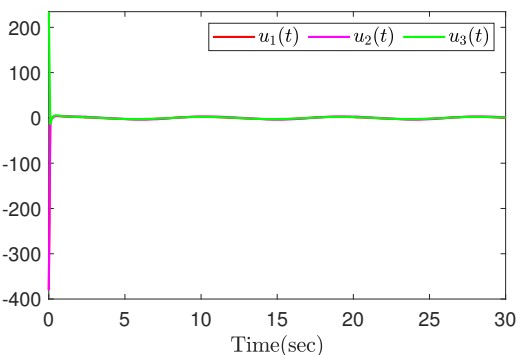

Fig. 5: Curves of optimal control input.

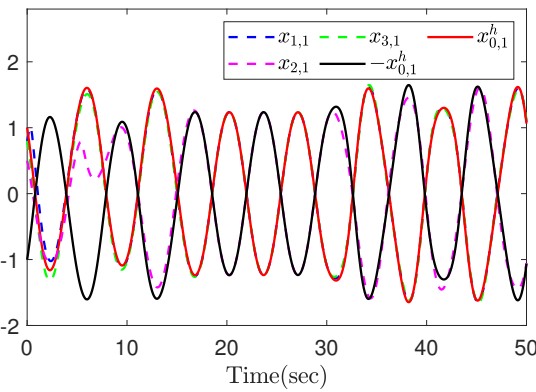

Fig. 2: Curves of $\tilde{x}_{i,1}$, $x_{0,1}^h$ and $-x_{0,1}^h$.

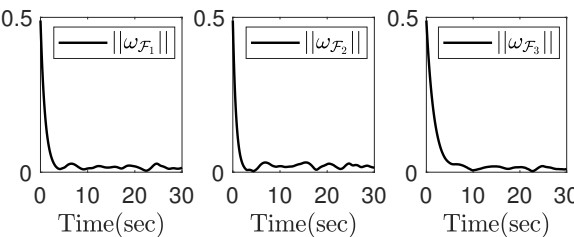

Fig. 6: Curves of $\|\omega_{\mathcal{F}_i}\|$.

From Fig. 2 and Fig. 3, the bipartite consensus can be achieved and the leader, followers 1 and 2 belong to a group while the follower 3 is geared to another group with opposite sign. Fig. 4 shows the bipartite consensus and the PT performance bounds. It can be obtained that the consensus error can reach the given accuracy 0.06 with the prescribed time $3s$. The optimal control input for each agent is depicted in Fig. 5, in which $u_i$ rapidly converges to a small region of zero. The norm of updating weights in unknown terms $\mathcal{F}_i(\mathscr{X}_i)$ are given in Fig. 6.

## VI. Conclusion

In this article, the problem of performance-based HiTL optimal bipartite consensus control for nonlinear MASs has been studied. First, the MASs have been monitored by human operator sending command signals to the non-autonomous leader to respond to any emergencies and guarantee the safety of MASs. Then, under the joint design architecture of prescribe-time performance function and error transformation, a novel performance index function has been developed to achieve optimal bipartite consensus with prescribed-time. Subsequently, the RL has been utilized to learn the solution to HJB equation, in which the FLSs are employed to implement the algorithm. The validity of the designed control scheme has been confirmed by simulation.

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
