# OpenReview forum: "Performance-Based Human-in-the-Loop Optimal Bipartite Consensus Control for Multi-Agent Systems via Reinforcement Learning"
_IEEE.org/ICIST/2024/Conference — IEEE ICIST 2024 Conference Submission_

### Official Review · Reviewer_R2cR · 2024-08-22

[review text omitted: it was posted to a different submission]

---

### Official Review · Reviewer_EmGi · 2024-08-22
**This article is quite fascinating and of high quality.**

**Rating:** 7
**Confidence:** 3

**Review:**

The paper titled "Performance-Based Human-in-the-Loop Optimal Bipartite Consensus Control for Multi-Agent Systems via Reinforcement Learning" investigates the performance-based human-in-the-loop optimal bipartite consensus control problem for nonlinear multi-agent systems under signed topology. Firstly, the MASs have been monitored by human operator sending command signals to the non-autonomous leader to respond to any emergencies and guarantee the safety of MASs. Then, under the joint design architecture of prescribe-time performance function and error transformation, a novel performance index function has been developed to achieve optimal bipartite consensus with prescribed-time. Finally, the RL has been utilized to learn the solution to HJB equation, in which the FLSs are employed to implement the algorithm. The article has clear logic and organization, my specific feedback is as follows: 1) In the introduction, the author’s analysis of the HiTL control is insufficient. 2) What role does the performance index function mentioned in the paper play in achieving the optimal bipartite consensus?

---

### Official Review · Reviewer_mWEK · 2024-08-28
**The obtained result is valuable and can be accepted after responding the following comments.**

**Rating:** 8
**Confidence:** 4

**Review:**

This paper  primarily investigates the design of an optimal bipartite consensus control scheme for nonlinear multi-agent systems (MASs) under signed topologies, with a human-in-the-loop (HiTL) component. The obtained result is valuable and can be accepted after responding the following comments.
1) In the introduction, it is not enough to state the current work. It should be expended and reconstructed. (2) In the simulation section, more analysis can be added to better explain the main results of this paper, that's not enough. (3) There are many typos and grammar errors. The authors should have a native English speaker or software packages to perform the editing check. (4) The conclusion of the article suggests using the present perfect tense for description

---

### Decision · Program_Chairs · 2024-09-06

Accept (Oral)